# Methyl Canthin-6-one-2-carboxylate Restrains the Migration/Invasion Properties of Fibroblast-like Synoviocytes by Suppressing the Hippo/YAP Signaling Pathway

**DOI:** 10.3390/ph16101440

**Published:** 2023-10-11

**Authors:** Zongying Zhang, Yunhan Wang, Qiuyun Xu, Xiaorong Zhou, Yong Ling, Jie Zhang, Liming Mao

**Affiliations:** 1Department of Immunology, School of Medicine, Nantong University, 19 Qixiu Road, Nantong 226001, China; 2113310013@stmail.ntu.edu.cn (Z.Z.);; 2Jiangsu Province Key Laboratory for Inflammation and Molecular Drug Target, School of Pharmacy, Nantong University, Nantong 226001, China; 3Basic Medical Research Center, School of Medicine, Nantong University, Nantong 226019, China

**Keywords:** methyl canthin-6-one-2-carboxylate, rheumatoid arthritis, MMP, fibroblast-like synoviocyte, Hippo/YAP

## Abstract

Rheumatoid arthritis (RA) is an inflammatory condition that causes severe cartilage degradation and synovial damage in the joints with multiple systemic implications. Previous studies have revealed that fibroblast-like synoviocytes (FLSs) play a pivotal role in the pathogenesis of RA. The appropriate regulation of FLS function is an efficient approach for the treatment of this disease. In the present study, we explored the effects of methyl canthin-6-one-2-carboxylate (Cant), a novel canthin-6-one alkaloid, on the function of FLSs. Our data showed that exposure to Cant significantly suppressed RA-FLS migration and invasion properties in a dose-dependent manner. Meanwhile, pre-treatment with Cant also had an inhibitory effect on the release of several pro-inflammatory cytokines, including IL-6 and IL-1β, as well as the production of MMP1 and MMP3, which are important mediators of FLS invasion. In further mechanistic studies, we found that Cant had an inhibitory effect on the Hippo/YAP signaling pathway. Treatment with Cant suppressed YAP expression and phosphorylation on serine 127 and serine 397 while enhancing LATS1 and MST1 levels, both being important upstream regulators of YAP. Moreover, YAP-specific siRNA or YAP inhibition significantly inhibited wound healing as well as the migration and invasion rate of FLS cells, an impact similar to Cant treatment. Meanwhile, the over-expression of YAP significantly reversed the Cant-induced decline in RA-FLS cell migration and invasion, indicating that YAP was required in the inhibitory effect of Cant on the migration and invasion of RA-FLS cells. Additionally, supplementation of MMP1, but not MMP3, in culture supernatants significantly reversed the inhibitory effect of Cant on RA-FLS cell invasion. Our data collectively demonstrated that Cant may suppress RA-FLS migration and invasion by inhibiting the production of MMP1 via inhibiting the YAP signaling pathway, suggesting a potential of Cant for the further development of anti-RA drugs.

## 1. Introduction

Rheumatoid arthritis (RA) is a prevalent inflammatory illness that causes increasing disability, systemic complications, early mortality, and significant socioeconomic expenses [1]. This disease affects approximately 1 in 200 individuals worldwide and manifests a considerably enhanced incidence in females. The etiology of RA has not been elucidated, while an increasing body of evidence suggests that many risk factors, such as a family history of RA, autoimmune illnesses, smoking, poor dental health, and viral infections, are linked to a higher chance of RA occurrence [2].

With the progression of technologies in early diagnosis and broader therapeutic choices, including the development of various disease-modifying antirheumatic medications (DMARDs) and biological therapies, the management and long-term prognosis of RA have been dramatically improved [3,4]. However, the defects of the conventional DMARDs such as poor solubility, rapid in vivo degradation, and low absorption rate, and the various side effects of the existing therapies limit the further progression of anti-RA therapies. Thus, the discovery of new anti-RA medicine with high efficacy and low toxicity is still an important and pressing issue in the research field of RA [5,6,7].

Even though RA causes systemic immune dysregulation and autoimmunity, the major clinical symptoms of RA are largely synovial inflammation and joint destruction [8]. The development of these symptoms is closely associated with the interplay of many cell types in the joints, including the fibroblast-like synoviocytes (FLSs) and the tissue resident macrophages [9]. Among these cells, FLSs have received increasing attention from recent studies of RA in recent years [10]. These cells manifest distinct aggressive characteristics and play an active role in disease development and progression [11,12]. The production of numerous matrix metalloproteinases (MMPs), including MMP1, MMP3, and MMP13, by FLS cells can destroy the structures of joint tissue, allowing FLSs to invade [10,13,14]. Meanwhile, FLSs also secrete several pro-inflammatory cytokines and chemokines, such as IL-1β, IL-6, and IL-8, which aggravate the pathological processes of RA [15,16]. Thus, appropriate regulation of FLS function has been suggested to be an effective approach in RA treatment.

The Hippo/YAP pathway has recently drawn increasing attention from RA researchers due to its potential regulatory role in the functions of RA-FLS cells [17]. The Hippo/YAP pathway is a highly conserved signaling cascade that exerts a key role in the control of tissue homeostasis and organ size by regulating cell proliferation, differentiation, and apoptosis [18]. The activity of this pathway depends on a sequential activation of a cascade of kinases, whereas its core components are the mammalian Ste20-like kinases1/2 (MST1/2) and large tumor suppressor 1/2 (LATS1/2). The upstream signals initiate the activation of the Hippo/YAP pathway by triggering the phosphorylation and activation of MST1/2 and its adaptor protein salvador 1 (SAV1). The interplay of MST1/2 and SAV1 subsequently induces the activation of LATS1/2 and the scaffold molecule Mps one binder 1 (MOB1), thereby inhibiting the downstream transcription co-activators, Yes-associated protein (YAP)/transcriptional coactivator with PDZ-binding motif (TAZ), from translocating to the nucleus and triggering the expression of target genes. Several precedent studies proposed that YAP signaling might affect the migration/invasion of FLS cells and might act as targets for treating RA [19,20], while the effects and associated mechanisms of this pathway on RA still need to be further verified.

Methyl canthin-6-one-2-carboxylate (Cant, Figure 1A) is a novel indole alkaloid derivative of canthin-6-one characterized in our recent study [21]. It can be found in a variety of plants, including *Zanthoxylum chiloperone*, *Aerva lanata*, *Eurycoma longifolia* roots, and *Simaba ferruginea A*. *St.-Hil* [22,23,24]. Previous studies showed that several canthin-6-one derivatives have a variety of biological activities [25], including a role in inhibiting inflammatory conditions in many diseases, such as diabetes and inflammatory bowel disease (IBD), via various mechanisms [26,27]. However, the role of Cant in inflammatory responses is unknown. In this study, we explored the impact of Cant on the functions of RA-FLS cells, a type of pathogenic cell involved in RA development. We found that Cant has a role in suppressing the migration/invasion properties and inhibiting the expression of pro-inflammatory mediators and MMPs by RA-FLS cells through the regulation of the Hippo/YAP signaling pathway. Our study suggested a potential of Cant as a candidate for further development of anti-RA drugs.

## 2. Results

### 2.1. Cant Did Not Increase Proliferation, Cell Cycle, or Apoptosis of RA-FLS Cells

Human RA-FLSs may influence the occurrence and progression of RA by invading the joint tissues and inducing cartilage damage. Before investigating the impacts of Cant on the functions of human RA-FLS cells, we first tested the effect of Cant on cell viability. To this end, we treated FLS cells with various doses of Cant for 24 or 48 h and then determined the cell viability using the CCK8 assay. As shown in Figure 1B, Cant treatment did not affect the cell viability of RA-FLS cells at the concentration of 40 μM for 24 h or 20 μM for 48 h. Based on these results, 20 μM or less of Cant was utilized in the subsequent experiments. To determine if Cant had a role in the apoptosis, cell cycle, and proliferation of RA-FLS cells, the cells were pre-treated with Cant in various concentrations and were then stained with annexin V-FITC/PI, PI, and EdU, respectively. The flow cytometry data revealed that the RA-FLS cells exposed to Cant did not undergo detectable apoptosis (Figure 1C,F, upper). In this experiment, the result of cells treated with methotrexate (MTX), an agent that can induce apoptosis of RA-FLS cells [28,29], was used as a positive control. Meanwhile, Cant also did not affect the cell cycle (Figure 1D,F, middle) or the proliferation rate of RA-FLS cells (Figure 1E,F, lower). These data demonstrated that Cant did not affect the apoptosis, cell cycle, or proliferation of RA-FLS cells.

### 2.2. Cant Suppressed Migration and Invasion of Human RA-FLS Cells

To determine the effects of Cant on the migration and invasion properties of RA-FLS cells, we pre-treated the cells with two doses of Cant and performed a transwell assay. The results showed that Cant treatment significantly suppressed both the migration and invasion of FLS cells in a dose-dependent manner (Figure 2A,C, upper). To further determine the effects of Cant on the migration of RA-FLS cells, we then exposed the cells to various doses of Cant and performed a wound-healing assay. The results showed that Cant exposure markedly suppressed the wound-healing rate of RA-FLS cells (Figure 2B,C, lower). Together, these data demonstrated that Cant had a role in suppressing the migration and invasion ability of human RA-FLS cells without destroying the cells.

### 2.3. Cant Suppressed the Expression of Pro-Inflammatory Cytokines and MMPs by RA-FLS Cells

The joint damage capability of human RA-FLS cells is associated with their ability to produce pro-inflammatory cytokines and MMPs [11]. To determine the potential effects of Cant on the expression of inflammation-associated cytokines by RA-FLSs, we exposed the cells to various doses of Cant and then stimulated the cells with TNF-α, a stimulator widely used in studies of RA to mimic the inflammatory microenvironment of the joint. The qPCR experiments showed that exposure to Cant suppressed the transcription of MMP1 and MMP3 (Figure 2D,E). Moreover, the levels of MMP1 and MMP3 in the culture supernatants also decreased after the administration of Cant (Figure 3A). Meanwhile, the transcription of the cytokines, including IL-6 and IL-1β, in FLS cells was also inhibited by Cant in a dose-dependent manner (Figure 2F,G). We examined the effect of Cant on the production of MMPs by RA-FLS cells using Western blot and found that pre-treatment with Cant markedly suppressed the protein expression of MMP1 and MMP3, but not MMP2 (Figure 2H–K). These data demonstrated that Cant may inhibit RA-FLS cell migration and invasion via selectively regulating the expressions of MMP1 and MMP3. Consistent with these results, we found that the levels of IL-6 in the culture supernatants of RA-FLS cells were significantly inhibited by the treatment with Cant (Figure 2L), while the level of IL-1β in the cell culture supernatants was not detectable. All these data indicated that Cant suppressed the production of both pro-inflammatory cytokines and MMPs.

### 2.4. Cant Suppressed Migration and Invasion of RA-FLS Cells by Attenuating MMP1 Expression

Previous studies revealed that the invasive ability of RA-FLS cells is associated with their production of MMPs. To clarify whether MMPs play a role in the effect of Cant on the inhibition of FLS invasion, as mentioned above, we examined the levels of MMP1 and MMP3 in the culture supernatants of FLS cells (Figure 3A) and thus determined the concentrations of MMPs used in subsequent studies. The RA-FLS cells were treated with recombinant human MMP1 (1.5 ng/mL) and/or MMP3 (57 ng/mL) to restore the levels in untreated RA-FLS cells. The cells were then treated with Cant (20 μM) for 24 h based on the lowered concentrations of MMP1 and MMP3 in the culture supernatants. In an invasion assay conducted using a transwell plate coated with matrix gel, we found that the supplementation of MMP1 or simultaneous supplementation of MMP1 plus MMP3 significantly reversed the inhibitory effect of Cant on RA-FLS cell vertical migration, while the supplementation of MMP3 alone did not have this effect (Figure 3B,D). In comparison, the supplementation of MMP1, MMP3, or MMP1 plus MMP3 all had no effect on the Cant-induced decline in RA-FLS cell vertical migration conducted using a transwell plate without coating matrix gel (Figure 3B,D). These results indicated that the inhibitory effect of Cant on the vertical migration of RA-FLS cells was mediated by MMP1, which might be required for RA-FLS cells to cross the matrix gel. In subsequent studies, we tested the effect of Cant on the horizontal migration of RA-FLS cells using a scratch test and found that the supplementation of MMP1 or MMP3 significantly impaired the effect of Cant on the inhibition of RA-FLS cell migration (Figure 3C,E). These data suggested a role of both MMP1 and MMP3 in promoting the horizontal migration of RA-FLS cells, possibly by regulating the cell adhesion and motility of the cells [30]. Together, these findings demonstrated that Cant may have inhibited the invasion of RA-FLS cells by suppressing the production of MMP1, while the Cant-induced decline in the horizontal migration of RA-FLS cells might have been associated with its inhibitory role in MMP1 and MMP3. 

### 2.5. Cant Suppressed the Expression and Activation of YAP/TAZ Pathway

The findings mentioned above led us to investigate the possible signaling pathways employed by Cant to achieve its impacts on RA-FLS cell functions. A previous study demonstrated that the migration/invasion properties of RA-FLSs are regulated by the YAP/TAZ pathway [20]. We thus asked if the YAP/TAZ pathway plays a role in the inhibition of Cant on the migration and invasion of RA-FLS cells. To this end, we first examined the effect of Cant on the expression level of YAP/TAZ in FLS cells using qPCR. The research demonstrated that Cant might suppress mRNA levels of important Hippo/YAP pathway indicators (Figure 4A). Compared with the untreated cells, the expression levels of YAP/TAZ in Cant-treated cells were markedly decreased (Figure 4A). Additionally, consistent with these observations, the upstream regulators of YAP, LATS1, and MST1 were found to be up-regulated by Cant stimulation (Figure 4A–C). Meanwhile, Cant stimulation also reduced the phosphorylation of YAP on serine 127 and serine 397 (Figure 4B,C), indicating a reduction in YAP activity. To support these findings, fluorescent microscopy immunofluorescence staining of Cant-treated FLSs revealed that Cant decreased YAP expression in FLSs in a dose-dependent manner (Figure 4D). Taken together, these findings demonstrated that Cant exerted an inhibitory role in the YAP/TAZ signaling pathway.

### 2.6. YAP Inhibition or Treatment with YAP-Specific siRNA Induced Phenotypes of RA-FLS Cells Similar to Treatment with Cant

To verify if the inhibition of the Hippo/YAP signaling pathway by Cant affected the migration and invasion of FLS cells, we transfected YAP-specific siRNA (si-YAP) into RA-FLS cells and examined RA-FLS functions. The results showed that si-YAP significantly reduced the expression of YAP (Figure 5A). More importantly, YAP knockdown by siRNA significantly suppressed the migration, as well as the invasion and wound-healing properties of RA-FLS cells, which were enhanced by stimulation of TNF-α (Figure 5B,C). Moreover, si-YAP suppressed both the TNF-α-induced production of pro-inflammatory cytokines and MMP1 and MMP3 (Figure 5D,E). Similarly, the ELISA results revealed that si-YAP inhibited the TNF-α-induced production of IL-6 by assaying cell supernatants (Figure 5F). These results suggested that the knockdown of YAP induced phenotypic changes in RA-FLS cells similar to Cant exposure, including decreased migration/invasion and reduced production of pro-inflammatory cytokines and MMPs. To further validate the role of YAP in regulating RA-FLS cell function, we next examined the effect of verteporfin (VP), a YAP inhibitor [31], on the mobility of RA-FLS cells. Similar to the results of siRNA-mediated knockdown of YAP, exposure to VP reduced both the transcription and protein expression of YAP (Figure 6A), while significantly inhibiting RA-FLS cell invasion and migration (Figure 6B,C) and the transcription levels of MMP1, MMP3, IL-6, and IL-1β (Figure 6D), which were all enhanced by TNF-α stimulation. Additionally, we also showed that VP treatment significantly suppressed the TNF-α-induced production of MMP1, MMP3, and IL-6 using Western blot or ELISA (Figure 6E,F). All these observations were phenotypically similar to those induced by the administration of Cant, indicating that the signals triggered by YAP might be involved in Cant-triggered decline in the functions of RA-FLS cells.

### 2.7. YAP Over-Expression Partially Reversed Cant-Induced Decline in RA-FLS Cell Migration and Invasion

To further validate whether the effect of Cant on inhibiting the migration and invasion of RA-FLS cells was mediated by inhibiting YAP, we over-expressed YAP in RA-FLS cells using a lentivirus-based transfection system. Using qRT-PCR and Western blot analysis, we showed that the expression of YAP was significantly elevated in terms of both mRNA and protein levels (Figure 7A). We then performed a transwell-based migration and invasion assay using these cells and found that the migration and invasion of RA-FLS cells with over-expressed YAP (OE-YAP) were significantly increased compared with those of the control cells (OE-NC; Figure 7B,C). Treatment with Cant still had an inhibitory effect on the migration and invasion of OE-YAP cells, while the inhibitory rate of Cant on these cells was much lower than that on control cells (Figure 7B,C). These observations provided further evidence that the inhibitory effect of Cant on the migration and invasion of RA-FLS cells was mediated by suppressing the expression of YAP. We then examined the effect of over-expressed YAP on expression of MMPs and pro-inflammatory cytokines by RA-FLS cells. The results showed that the transcriptions of MMP1, MMP3, IL-6, and IL-1β in OE-YAP cells were much higher than those in OE-NC cells (Figure 7D–G). Consistent with these findings, the protein levels of MMP1, MMP3 (Figure 7H–J), and IL-6 (Figure 7K) were also significantly increased in OE-YAP cells compared with those in OE-NC cells. The level of IL-1β was undetectable in ELISA assay. It should be noted that the inhibitory effect of Cant on the protein levels of MMPs was remarkably reduced in OE-YAP cells (Figure 7H–J), a result consistent with the above observations. However, the transcription levels of the MMPs and cytokines in Cant-treated OE-YAP cells were comparable with those in Cant-treated OE-NC cells (Figure 7D–G). Cant stimulation could, to some extent, down-regulate YAP levels in OE-YAP cells, but the inhibitory effect was reduced when compared with that in OE-NC cells (Figure 7L–M). These data suggested that the presence of high levels of YAP did not affect the inhibitory effect of Cant on the transcription of MMPs and pro-inflammatory cytokines, but indeed impaired Cant’s inhibition on MMP protein levels, demonstrating that YAP was required in the inhibitory effect of Cant on MMP proteins, but was dispensable for the inhibition of Cant on the transcription of MMPs and the cytokines. Taken together, our data demonstrated that Cant could inhibit YAP and thus down-regulate the protein levels of MMPs and prevent the migration and invasion of RA-FLS cells. Meanwhile, the inhibitory effect of Cant on the transcriptions of MMPs and pro-inflammatory cytokines was independent of YAP.

## 3. Discussion

In this study, we reported the impacts of Cant on suppressing the migratory and invasive properties of FLSs from RA patients. In further studies, we showed that Cant inhibited the production of pro-inflammatory cytokines and MMPs. The compensation of MMP1, but not MMP3, attenuated the inhibitory effect of Cant on the migration and invasion of RA-FLSs. For the mechanisms, using siRNA-mediated knockdown, an inhibitor of YAP, and a lentiviral-vector-mediated over-expression of YAP, we showed that the inhibitory effect of Cant on the functions of RA-FLS cells was achieved, at least in part, by suppressing the Hippo/YAP signaling pathway. Our data suggested that Cant might have a promising application for the further development of anti-RA drugs.

RA is an inflammatory condition that causes pannus formation and cartilage damage due to over-proliferation of the synovium. Although significant progress in basic and clinical RA research has been achieved in recent years, the management and treatment of this disease has still not been satisfactory and its prevalence is increasing. According to recent reports, more than 2% of individuals over 60 may be affected by RA [32]. Although the exact etiology of RA is unclear, some precedent studies have proven that the development or progression of RA is associated with the imbalanced regulation of immune responses due to genetic defects or influences of particular invading microbes, which causes the aberrant activation of various inflammatory cells in the joints, such as macrophages, lymphocytes, neutrophils, and FLS cells. Thus, how to effectively keep the activity of various immune cells as well as the levels of numerous inflammatory mediators in check is the key to the treatment of RA. With the rapid progression of the understanding of RA pathogenesis, many highly effective agents have been developed and applied to RA as first-line treatments, including numerous biological agents and targeted synthetic DMARDs [33], which have largely improved the prognosis of RA patients. However, drug resistance, the high cost, and increasing risk of side effects, such as serious infections and cancers, occur in a large proportion of RA patients, causing refractory inflammatory attack on the joints. Refractory RA is currently defined arbitrarily, and there is little information available on its outcomes and effects on such individuals [34,35]. These observations limit the application of biological agents and DMARDs in RA therapy. Therefore, it is necessary to find novel treatment agents that are both efficient and secure. 

The identification of natural compounds derived from a range of herbs and other natural sources has drawn increasing attention from RA researchers and has benefited the development of anti-arthritic medicine during the past few decades [36]. Cant is a novel canthin-6-one alkaloid belonging to a subclass of β-carbolines with one extra D-ring [37]. This class of alkaloids has multiple biological activities, including antiviral [38], antibacterial [39,40], antifungal [41,42], and antitumor effects [35,43,44]. Intriguingly, several precedent studies proved that several canthin-6-one alkaloids can suppress the expression of many inflammatory mediators such as NO and PGE2 by macrophages and astrocytes, and thus have anti-inflammatory roles in multiple animal models of human diseases [45]. However, as a novel canthin-6-one alkaloid, the role of Cant in inflammation and inflammatory responses has not been studied. Our current study explored the effects of Cant on the RA-associated pathogenic cells, RA-FLS cells. We provided evidence that Cant had a role in suppressing the activation and function of FLS cells obtained from RA patients. It also suppressed the production of pro-inflammatory cytokines and MMPs. These findings remind us of a previous report by Fan et al. [27], showing that the oral administration of another Cant derivative, 4-methoxy-5-hydroxycanthin-6-one, might block experimental arthritis in rats induced by carrageenan or complete Freund’s adjuvant (CFA). Our data, together with previous studies, suggested the role of Cant or its derivative in suppressing the inflammatory response in RA and may have a potential application in therapeutic intervention against RA. In further studies, we will perform in vivo experiments to elucidate whether Cant has a role in animal models of arthritis. 

RA-FLS cells play a prominent role in the development of RA by largely invading the extracellular matrix and producing cytokines and MMPs, which are crucial in the deterioration of joint symptoms in RA. Invasive cells dynamically reorganize their actin cytoskeletons and regulate the formation of protrusive structures and provide the pressures necessary for cell translocation. The steadily activated RA-FLSs have an aberrant potential for migration [11,46,47]. Accordingly, the modulation of RA-FLS migration and invasion has been proven to be effective in ameliorating RA-related tissue damage. We thus investigated the impact of Cant on the migration/invasion properties of RA-FLSs in this study. We observed that Cant had an suppressive effect on the migration/invasion of the cells. These data imply that Cant might block RA-FLS invasion to the synovium and the consequent joint degeneration, thereby ameliorating symptoms in RA patients. 

MMPs are implicated in the etiology of RA by degrading cartilage and are useful indicators for predicting joint function and imaging outcomes in RA. MMPs in the serum are correlated with disease activity and RA joint deterioration progression [48,49,50]. By Western blot, ELISA, and qRT-PCR analysis, we found that Cant treatment dramatically reduced the production of MMP1 and MMP3, but not MMP2, in TNF-α-stimulated RA-FLSs, indicating that one of the ways by which Cant regulates the aggressive behavior of RA-FLSs is by reducing their production of MMP1 and MMP3. It should be noted that the production of MMP2 was not affected by Cant. This phenomenon may be explained by the diversity in the regulating pathways of different MMPs. Cant may stimulate the expression of particular regulatory molecules, such as various tissue inhibitors of MMPs, which selectively regulate the expression of some MMPs, but do not affect others [51]. Further studies need to identify the differential regulatory molecules or pathways that may be affected by Cant, which may differentially affect the production of MMPs tested in our study. Subsequently, we employed a compensation study to verify the effects of MMPs on the migration and invasion of RA-FLSs. We found that the inhibitory effect of Cant on the invasion of RA-FLSs was mediated by MMP1, but not MMP3. This observation was different from the findings by Ma et al. [29], showing that the artesunate-induced inhibition of RA-FLS invasion was mediated by suppressing MMP9. These data imply that different compounds may target various MMPs to regulate the migration and invasion of RA-FLSs. In our study, MMP1 might be helpful for RA-FLS cells in crossing the matrix gel, while MMPs might regulate numerous signaling pathways and thus modulate cell motility and cell adhesion, which may also contribute to the invasion activity of RA-FLS cells [30]. Our study also revealed a different effect of MMPs on the restoration of the reduced horizontal and vertical migration of RA-FLS cells triggered by Cant. This observation might be explained by the possible effect of different MMPs in regulating rearrangement of cytoskeletal elements, which might affect the migrating features of RA-FLS cells. This speculation needs to be verified in future studies. 

Our study provides evidence that MMPs may be regulated by Hippo/YAP signaling; however, this result cannot exclude the possible involvement of other signaling pathways in the inhibitory role of Cant, such as the NF-κB, MAPK, or STAT3 pathways, which have been previously implicated in the anti-inflammatory properties of many canthin-6-one alkaloids [52,53]. As described above, the Hippo/YAP pathway has been reported to play a role in regulating the migration and invasion of RA-FLS cells [20]. The potential effects of Cant or other canthin-6-one alkaloids on the functions of RA-FLS cells have not been studied. On this basis, we designed this study and found that the Hippo/YAP signaling pathway indeed played a critical role in the Cant-triggered inhibition of RA-FLS cell functions. However, as a family member of the canthin-6-one alkaloids, it is reasonable to assume that Cant may also affect the function of many other signaling pathways involved in regulating RA-FLS cell functions. Further systematic investigation of the participation of numerous pathways in the inhibitory role of Cant on RA-FLS cells may expand our understanding of the overall molecular mechanisms of the anti-inflammatory properties of this compound and may facilitate the development of new anti-RA drugs based on the application of Cant. 

One of the most prevalent manifestations in RA patients is inflammation-mediated damages in synovial tissue of the joints, while the over-production of multiple cytokines, including TNF-α, IL-1β, and IL-6, plays a central role in this process [54]. Thus, the manipulation of the inflammatory response and the levels of cytokines in the joints is a widely used approach in RA management [33]. Our data revealed the role of Cant in suppressing cytokine production, including IL-6 and IL-1β, by RA-FLS cells. Moreover, the alteration in cytokine levels produced by RA-FLSs during exposure to Cant was accompanied by the expression levels of YAP, indicating a role of the YAP signaling pathway in the modulation of inflammatory cytokines. These data, together with many previous reports, confirmed the role of the YAP signaling pathway in regulating pro-inflammatory cytokines. While similar to the impact of YAP on MMPs, the exact regulatory mechanism of this pathway in regulating cytokine production still requires further investigation. We also examined the possible role of Cant in apoptosis and found that treatment with Cant at the concentration of 40 μM or less did not affect the apoptosis of RA-FLS cells. Thus, its role was different to that of the YAP inhibitor, VP, which suppresses YAP/TAZ transcription and reduces the resistance of RA-FLS cells to apoptosis [17]. Therefore, the effect of VP on RA-FLS cells may be partially mediated by facilitating cell apoptosis. In our study, although the activity of the YAP signaling pathway was also suppressed by treatment with Cant, its effect was not achieved by inducing apoptosis of the cells, but by suppressing migration and invasion, as well as the production of pro-inflammatory cytokines and MMPs of RA-FLS cells. 

Of note, our data obtained using cells over-expressing YAP disclosed that the effects of Cant on the functions of RA-FLS cells was not always dependent on YAP. When treated with Cant, the migration and invasion of cells over-expressing YAP (OE-YAP) were much lower than those of cells expressing normal levels of YAP (OE-NC). Consistent with this, the inhibitory effect of Cant on the protein levels of MMP1 and MMP3 was significantly reduced in OE-YAP cells. These findings suggested that the inhibitory effect of Cant on the migration and invasion of RA-FLS cells, and the protein levels of MMPs, was mediated by suppressing the expression of YAP. However, the transcription levels of the MMPs and cytokines in OE-YAP cells were not affected by the over-expression of YAP when compared with those in OE-NC cells after treatment with Cant. These observations indicated that the over-expression of YAP did not change the transcription of MMPs and pro-inflammatory cytokines, but indeed enhanced the protein levels of MMPs. These findings provided evidence that YAP was involved in the inhibition of MMP proteins, but was not required in the transcription of the MMPs and cytokines during treatment with Cant. Our data suggested further complexity to the role of the YAP signaling pathway in regulating gene expression. The exact regulatory network of YAP in modulating functions of RA-FLS cells, especially the transcription and translation of MMPs, needs to be further elucidated in future studies. 

## 4. Materials and Methods

### 4.1. Materials and Reagents

Methyl canthin-6-one-2-carboxylate (Cant) was prepared according to the literature [21]. YAP inhibitor Verteporfin was obtained from Sigma-Aldrich (SML0534-5MG, St. Louis, MO, USA). The human recombinant TNF-α was purchased from Pepro Tech (300-01A-2UG, Rocky Hill, NJ, USA). Human MMP1 (420-01-2UG, Rocky Hill, NJ, USA) and MMP3 recombinant proteins were all obtained from Pepro Tech (420-01-2UG, Rocky Hill, NJ, USA).

### 4.2. Cell Culture

The immortalized human RA-FLS cells were purchased from Delf Biotechnology Co., Ltd. (tings-951129, Hefei, China). The cells were cultured in Dulbecco’s modified Eagle’s medium (DMEM, C11995500BT, Gibco, Grand Island, NY, USA) containing 10% fetal bovine serum (FBS, 10270-106, Gibco, USA), 2 mM L-glutamine, 50 g/mL gentamicin, 100 units/mL penicillin, and 100 m/L streptomycin at 37 °C in a humid environment containing 5% CO_2_.

### 4.3. CCK8 Assay

The same culture conditions as previously were used to determine whether cells were viable using a cell-counting kit (CCK8 kit, abs50003-5ml, Absin, Shanghai, China). The experiment consisted of four groups: a control group (cells + medium), a blank group (medium), and a group with different drug delivery concentrations (cells + medium + 10/20 μM Cant). Cells were counted, adjusted to a concentration of 2 × 10^5^ mL, and planted at 100 μL/well in 96-well plates using cells from the various treatment groups. Three copies of the cells from each treatment group were planted. The plates were placed in an incubator (37 °C and 5% CO_2_) and the cells were incubated until the appropriate time. Then, the drug for the pair was added to the corresponding wells at the set concentration. CCK8 solution (10 μL) was added to each well. Then, the cell culture plates were allowed to sit in the incubator for one to four hours. Finally, a plate reader was used to measure the absorbance at 450 nm. Untreated cells, medium, and CCK8 solution were also tested as control wells.

### 4.4. Transwell Migration Assays

The upper chamber of the Transwell (354480, Biocoat, Corning, NY, USA) was infected with 5.0 × 10^4^ cells from the control or treatment groups, and the cells were left to migrate for 24 h in chemoattractant-rich media (DMEM + 5% FBS) at 37 °C. Following incubation, the cells were fixed with paraformaldehyde (4%) for 5 min and then were stained with crystal violet (0.2%, C0121, Beyotime, Shanghai, China) for 30 min. To count the cells moving through each transwell, four photos were captured using a microscope. For each circumstance, three chambers were counted.

### 4.5. Invasion Assays

First, 5.0 × 10^4^ cells from the control or treatment groups were planted into the upper chamber of the transwell upper chamber (354480, Biocoat, USA) and allowed to colonize for four days in the chemosynthetic-rich medium (DMEM + 5% FBS) below after a sufficient amount of matrix gel (356231, BD, San Diego, CA, USA) had been dispersed throughout and had had time to harden. Following incubation, cells were fixed with 4% paraformaldehyde for 5 min at room temperature and then were stained with 0.2% crystal violet (C0121, Beyotime, Shanghai, China) for 30 min. To count the cells in the infected chamber, four photos were captured using a microscope for each chamber. For each condition, three chambers were counted.

### 4.6. Wound-Healing Assays

We measured the impact of the control or treatment group cell migration using the scratch wound-healing assay. In 6-well plates, RA-FLS cells (2 × 10^5^/well) were cultured in DMEM containing 10% FBS at 37 °C for 24 h in a CO_2_ incubator. After setting up a blank control, scratches were made at the bottom of each well using a p200 pipette tip, and cells were then treated with Cant (10 Μm/20 μM) or VP (0.5 μM/1 μM) or YAP siRNA. At 0, 24, and 48 h after scratching, representative photos were taken under a microscope, and Image J software (Version 1.47, National Institutes of Health, Bethesda, MD, USA) was employed to quantify and evaluate the scratched regions.

### 4.7. Western Blot

The treated RA-FLS cells were rinsed with room temperature PBS before being lysed in 1% NP40 buffer. Before adding Loading, the extracted proteins were measured and the concentrations were uniformly adjusted before turning on the thermostatic heater to 99 °C and cooking the protein samples for 10 min at 99 °C. Each sample was resolved on a 4–15% SDS-polyacrylamide gel (P0466M, Beyotime, Shanghai, China) and transferred to NC membranes (abs959, Absin, Shanghai, China) using the semi-dry transfer (Bio-Rad Laboratories GmbH, München, Germany) procedure. Place NC membranes in 5% skim milk (P0216, Beyotime, Shanghai, China) powder and mix for 1-2 h at room temperature. Membranes were blotted with primary antibodies recognizing MST1 (14946, Cell Signaling Technology, Boston, MA, USA), LATS1 (3477S, Cell Signaling Technology, USA), YAP (14074T, Cell Signaling Technology, USA), TAZ (8418S, Cell Signaling Technology, Boston, USA), P-YAP397 (13619T, Cell Signaling Technology, USA), P-YAP127 (13008T, Cell Signaling Technology, USA), MMP1 (54376S, Cell Signaling Technology, USA), MMP3 (14351S, Cell Signaling Technology, USA) and GAPDH (AF0006, Beyotime, Shanghai, China), respectively, and followed with horseradish peroxidase (HRP)-conjugated secondary antibodies (Beyotime, Shanghai, China). In the WesternBright^TM^ Sirius (K-12043-D10, Menlo Park, CA, USA), combine liquids A and B in a 1:1 ratio and configure the necessary amount of developer using the Tanon gel imager.

### 4.8. ELISA

RA-FLS cells were pre-treated with Cant for 1 h and then were stimulated with TNF-α (50 ng/mL) for 24 h. The levels of IL-6 (555220, BD Biosciences, Beijing, China), MMP1 (MK0072B, Mekebio, Shanghai, China), and MMP3 (MK00108B, Mekebio, Shanghai, China) were determined using commercial ELISA kits following manufacturer’s instructions.

### 4.9. Quantitative Polymerase Chain Reaction (qPCR)

Total cellular RNA of RA-FLS cells was extracted using RNeasy Mini Kit (74104, Qiagen, MD, USA). Complementary DNA (cDNA) samples were synthesized using the RevertAid First Strand cDNA Synthesis kit (K1622, Thermofisher, Waltham, MA, USA). Then, cDNA was amplified in accordance with the SYBR Green RT-PCR reaction kit’s instructions (A25742, Thermofisher, Waltham, MA, USA). Table 1 lists the primers used for qRT-PCR analysis. The results of the test were calculated by relative quantitative analysis of 2^−ΔΔCT^. All tests were performed three times.

### 4.10. SiRNA-Mediated Knockdown of YAP

The siRNA for YAP and negative control (NC) was synthesized by Shanghai GENEray (Shanghai, China). Three independent siRNAs were designed: siRNA#1: (5′-GGUCAGAGAUACUUCUUAATT-3′; 5-UUAAGAAGUAUCUCUGACCTT-3′), siRNA#2: (5-GGAGAAAUUUACUAUAUAATT-3′; 5′-UUAUAUAGUAAAUUUCUCCTT-3′, and siRNA#3: (5′-GGUGAUACUAUCAACCAAATT-3′; 5′-UUUGGUUGAUAGUAUCACCTT-3′). The siRNA transfection procedures were performed according to ribo FECT TMCP Transfection Kit (C10511, RiboBio, Guangzhou, China). Total RNA of siRNA-treated RA-FLS cells was extracted to assess knockdown efficiency of YAP by qPCR. The siRNA sequence with the most silencing efficiency was used in subsequent studies. Nonspecific NC siRNA were also designed and synthesized. The mock groups was those treated with the transfection reagent only.

### 4.11. Cell Apoptosis Assay

The effects of Cant on apoptosis of RA-FLS cells were determined using flow cytometry. Briefly, a total of 1 × 10^5^ cells per well were seeded into 6-well plates and incubated with various doses of Cant (10 and 20 μM) for 24 h. The cells were then collected and rinsed three times in PBS. For apoptosis assay, the cells were incubated with Annexin V-FITC and PI (556547, BD, San Diego, CA, USA) for 20 min at room temperature. Cell cycle and apoptosis were quantified using the Beckman flow cytometer. According to the manufacturer’s instruction, the untreated control cells were FITC Annexin V-FITC and PI double negative (lower left), indicating that these cells were primarily viable and not undergoing apoptosis; although the gating strategy had excluded most of the cell debris, some residue debris or dead cells were still detectable (upper left); In this experiment, we used MTX-treated cells as a positive control. Cells undergoing apoptosis were Annexin V-FITC positive and PI negative (lower right); The cells of Annexin V-FITC and PI double positive cells were in end stage apoptosis (upper right). 

### 4.12. Immunofluorescence Analysis

RA-FLS cells were cultivated in a sterile 24-well plate and treated with Cant (10 and 20 μM) for 24 h. The cells were then washed three times with PBS and fixed with 4% paraformaldehyde in PBS for 30 min. Subsequently, a penetration step was performed with 0.5% TritonX-100 for 15 min. Then, the cells were rinsed three times with PBS and sealed with 2% BSA for 1 h. The cells were incubated with YAP-specific antibody (14074T, Cell Signaling Technology; 1:100, Shanghai, China) at 4 degrees overnight. Then, the cells were washed three times with PBS and incubated with Rhodamine coupled goat anti-Rabbit IgG (abcam, Shanghai, China) in the dark at room temperature for 2 h. The nuclei were transstained with 0.5 μg/mL DAPI (ab104139, Sigma-Aldrich, St. Louis, MO, USA). Fluorescence signal was captured using a fluorescence microscope.

### 4.13. EdU Proliferation Assay

Cells were treated for 24 h with the appropriate concentrations of Cant before being plated at a density of 5 × 10^4^ cells/mL into 24-well culture plates. The EdU staining was performed according to the manufacturer’s procedure using a Cell-Light EdU DNA Cell Proliferation Kit (KGA331-100, KeyGEN BioTECH, Nanjing, China). The cells were then treated with EdU (1:1000) for 8 h before being washed, rinsed in PBS, fixed with paraformaldehyde, and permeabilized for 10 min with 0.3% Triton X-100 in PBS. Cells were treated in the dark for 30 min with the Apollo staining reaction solution. Following that, cells were nuclear stained for 5 min with DAPI and pictures were acquired using a fluorescence microscope.

### 4.14. Cell-Cycle Analysis

Flow cytometry was employed to detect the effects of Cant on the cell cycle of RA-FLS. In brief, 1 × 10^5^ cells were planted onto 6-well culture plates and cultured with Cant for 24 h. After that, cells were collected and rinsed three times in PBS. The cells were subsequently resuspended in 70% pre-chilled ethanol and fixed at 20 °C overnight for cell-cycle analysis. The cells were rinsed again with PBS and resuspended in 200 L PI/RNase Staining Buffer (550825, BD Pharmingen, San Diego, CA, USA) for 30 min before flow cytometric analysis.

### 4.15. Over-Expression of YAP in RA-FLS Cells

The packaged poSLenti-EF1-EGFP-P2A-pour-GMV-MCS-3xFLAG-WPRE lentiviral vector carrying human YAP or the empty vector was transfected into RA-FLS cells, and the medium was discarded after 14 h and replaced with fresh DMEM. The cells were stabilized for 72 h and a puromycin-based screening was performed for two weeks. The expression of YAP was tested in the resultant cell lines using qPCR and Western blot assays.

### 4.16. Statistical Analysis

All procedures of statistical analyses in the study were conducted using GraphPad Prism 8.0 software (La Jolla, CA, USA). The differences between multiple groups were analyzed using ANOVA. A significance threshold of *p* < 0.05 was applied to determine statistical significance. 

## 5. Conclusions

In summary, our study investigated the effects of a novel canthin-6-one alkaloid, Cant, on the functions of RA-FLS cells. The compound was generated and characterized in our recent study [21], but its role in inflammation is unclear. Our data provided evidence that Cant could suppress the pro-inflammatory functions of RA-FLS cells, including migration, invasion, and the production of MMPs and pro-inflammatory cytokines. For the mechanisms, the suppressive role of Cant was achieved by inhibiting the Hippo/YAP signaling pathway (Diagram in Figure 8). To our knowledge, this is the first study describing the effect of a canthin-6-one family of compounds in RA-FLS cells. The suppressive effect of Cant on the Hippo/YAP signaling pathway has also not been reported in the literature. Our findings collectively imply that Cant may be a promising candidate for the further development of RA-treating drugs by targeting the inflammatory functions of RA-FLS cells.

## Figures and Tables

**Figure 1 pharmaceuticals-16-01440-f001:**
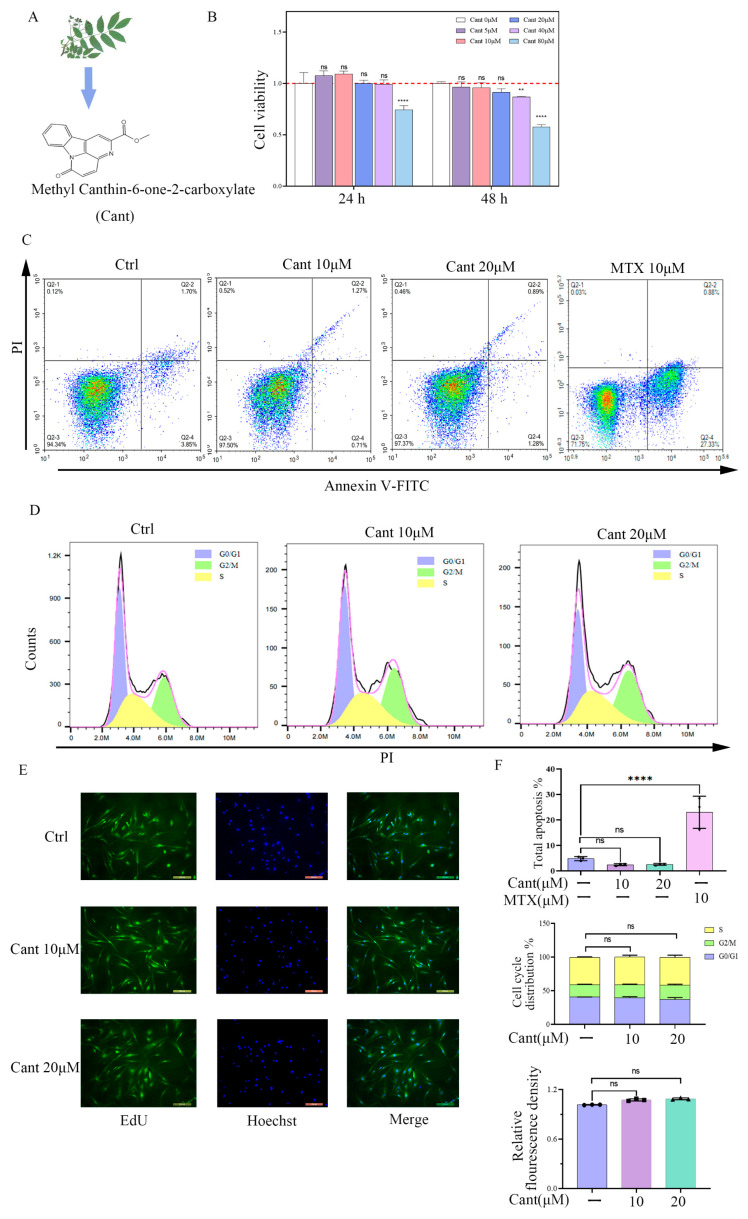
Cant does not increase proliferation, cell cycle, or apoptosis of RA-FLS cells. (**A**) Chemical formula of canthin-6-one. (**B**) RA-FLS cells were incubated with various doses of Cant (0, 10, and 20 μM) for 24 h or 48 h. Then, CCK8 assay was performed to examine the impact of Cant on cell proliferation. (**C**) Annexin V-FITC/PI double staining and flow cytometry were used to determine the status of apoptosis. The result of MTX-treated cells was used as a positive control. (**D**) Cell cycle was detected by flow cytometry with PI staining. (**E**) EdU tests were used to assess the status of RA-FLS cell proliferation. Representative photographs of RA-FLS cells stained with EdU (green) and DAPI (blue, ×100) were shown. (**F**) Quantified data of the effects of Cant on apoptosis, cell cycle, and proliferation of RA-FLS determined by annexin V-FITC/PI, PI, and EdU, respectively. All experiments were performed independently three times. The data are shown as mean ± SEM. ** *p* < 0.01, **** *p* < 0.0001, in comparison to RA-FLS without Cant treatment.

**Figure 2 pharmaceuticals-16-01440-f002:**
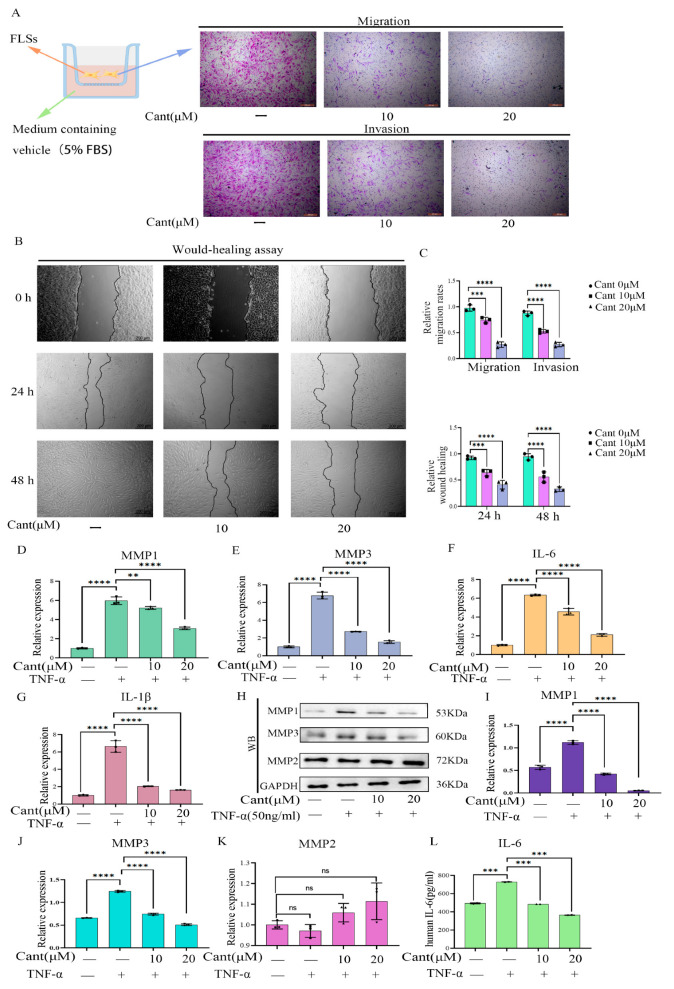
Cant inhibited migration/invasion, and the production of MMPs and pro-inflammatory cytokines by RA-FLS cells. (**A**,**B**) The effects of different concentrations of Cant on RA-FLS cell migration/invasion were determined by transwell (**A**) and wound-healing assay (**B**), respectively. The displayed images are representatives from three independent experiments (×100). (**C**) The migrated or invaded cells in five random fields of each replicate (upper) were quantified, and the statistical chart of the wound-healing results (lower) is shown. The effect of Cant on the transcription of MMP1 (**D**), MMP3 (**E**), IL-6 (**F**), and IL-1β (**G**) induced by TNF-α (50 ng/mL) by RA-FLSs was determined by qRT-PCR. (**H**) The levels of MMP1 and MMP3 in Cant-stimulated RA-FLSs were detected by Western blot analysis. The intensity values of the bands of MMP1 (**I**), MMP3 (**J**), and MMP2 (**K**) in (**H**) were quantified and subjected to statistical analysis. (**L**) The release of IL-6 in the supernatant of Cant-treated cells was detected by ELISA. All experiments were conducted independently three times. The data are shown as mean ± SEM. *** *p* < 0.001, **** *p* < 0.0001, in comparison to RA-FLS cells without TNF-α or Cant treatment (**C**). ** *p* < 0.01, *** *p* < 0.001, **** *p* < 0.0001, compared with RA-FLS treated with TNF-α alone (**D**–**L**).

**Figure 3 pharmaceuticals-16-01440-f003:**
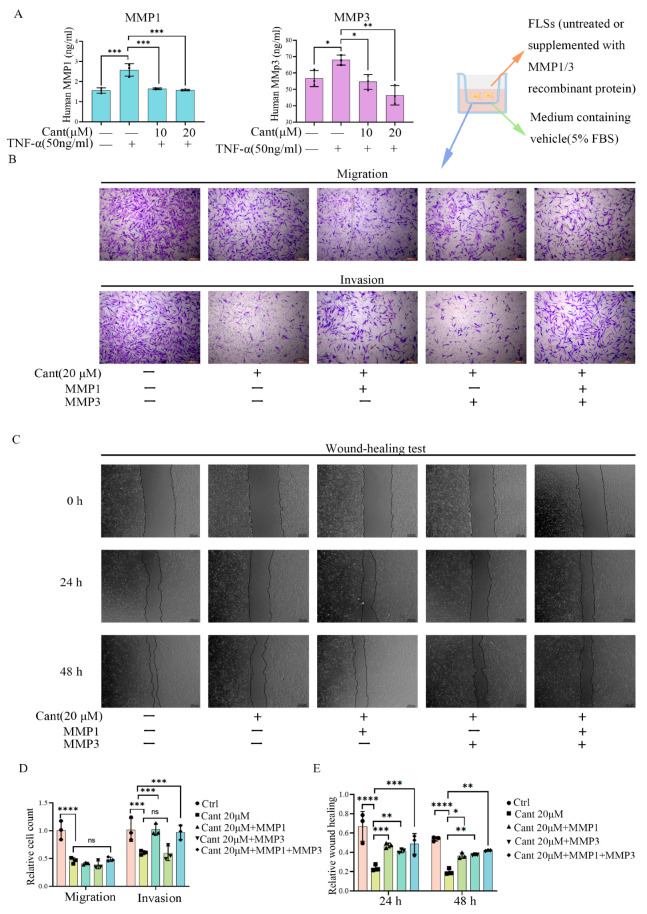
MMP1 was involved in the effects of Cant on inhibiting RA-FLS cell migration and invasion. (**A**) The contents of MMP1 and MMP3 in the supernatant of Cant-treated (10, 20 μM) RA-FLS cells were detected by ELISA. RA-FLS cells were treated with recombination human MMP1 (1.5 ng/mL) and/or MMP3 (25 ng/mL), the cells were then treated with Cant (20 μM) for 24 h and were subjected to transwell assay (**B**) and wound-healing assay (**C**). The migrated or invaded cells in five random fields of each experimental replicate were quantified (**D**). The statistical chart of the wound-healing results is shown (**E**). All the experiments were conducted independently three times. The data are shown as mean ± SEM. * *p* < 0.05, ** *p* < 0.01, *** *p* < 0.001, in comparison to RA-FLS treated with TNF-α alone (**A**). * *p* < 0.05, ** *p* < 0.01, *** *p* < 0.001, **** *p* < 0.0001, in comparison to RA-FLS without TNF-α or Cant treatment (**D**,**E**).

**Figure 4 pharmaceuticals-16-01440-f004:**
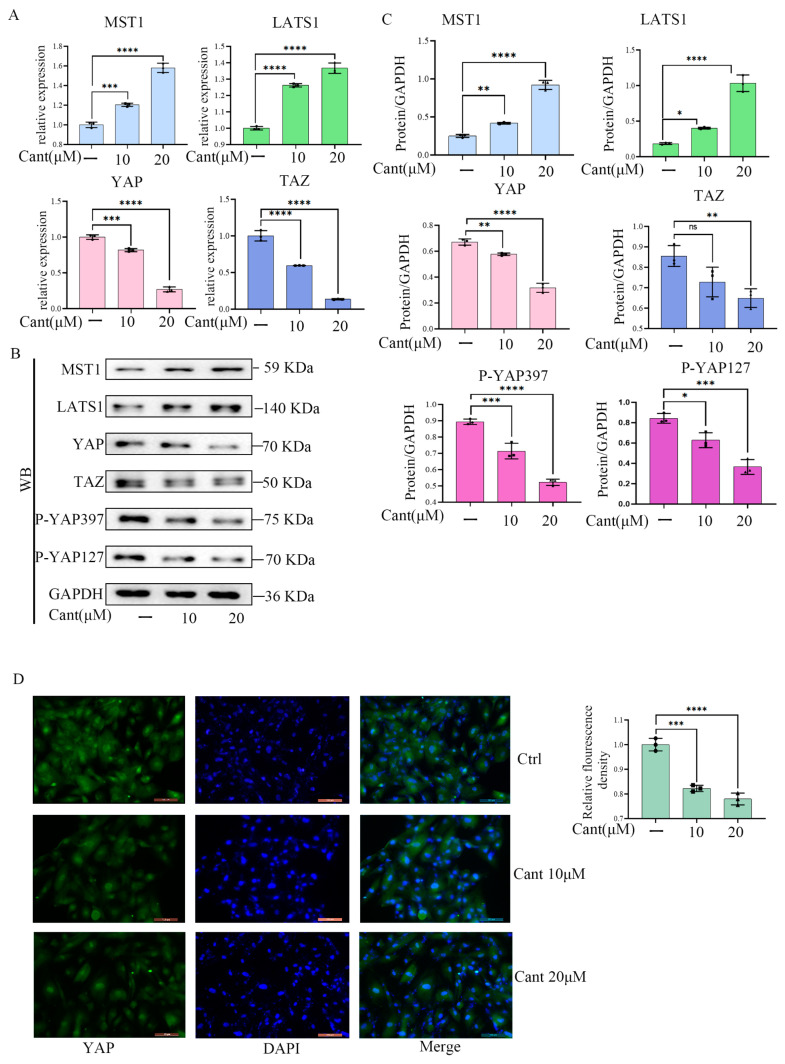
Cant suppressed the expression of YAP/TAZ. (**A**) RA-FLSs were treated by Cant (10, 20 μM) to detect MST1, LATS1, YAP, and TAZ mRNA levels. (**B**,**C**) RA-FLS cells were pre-treated with Cant (10, 20 μM) for 24 h. Then, cells were harvested, and the cell lysates were subjected to Western blot to detect the expression of MST1, LATS1, YAP, TAZ, P-YAP397, and P-YAP127. (**D**) YAP detected by immunofluorescence staining assay in Cant-treated RA-FLS cells. Magnification, ×20. All the experiments were conducted independently three times. The data are shown as mean ± SEM. * *p* < 0.05, ** *p* < 0.01, *** *p* < 0.001, **** *p* < 0.0001, in comparison to RA-FLSs without Cant treatment.

**Figure 5 pharmaceuticals-16-01440-f005:**
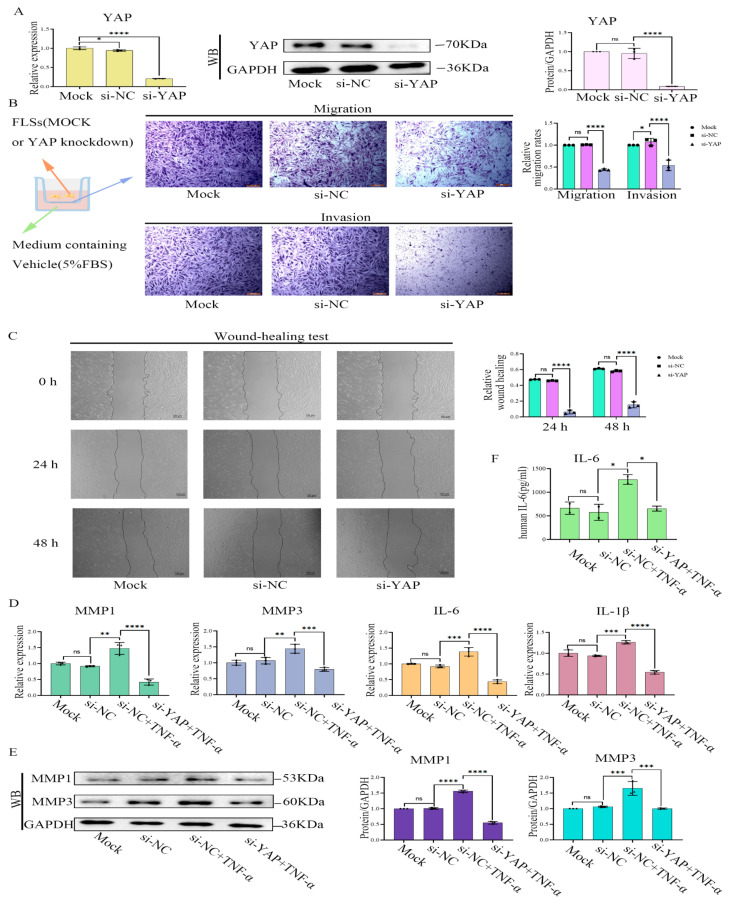
The treatment with YAP-specific siRNA resulted in reduced inflammatory functions of RA-FLS cells. (**A**) The effect of si-YAP on the expression of YAP in RA-FLS was measured by Western blot and qPCR. (**B**,**C**) Transwell assays and wound-healing tests showed the capacity of the vertical migration, invasion (**B**), and horizontal migration (**C**) of si-YAP-treated RA-FLS cells. (**D**,**F**) The effect of si-YAP on the levels of MMP1, MMP3, IL-6, and IL-1β in RA-FLS were measured by qPCR (**D**), and IL-6 in the cell culture supernatant was tested by ELISA (**F**). (**E**) Forty-eight hours after si-YAP siRNA transfection, the cells were treated with TNF-α (50 ng/mL). Then, the levels of MMP1 and MMP3 were analyzed by Western blot. All experiments were conducted independently three times. The data are shown as mean ± SEM. * *p* < 0.05, **** *p* < 0.0001, in comparison to si-NC RA-FLS group (**A**–**C**). * *p* < 0.05, ** *p* < 0.01, *** *p* < 0.001, **** *p* < 0.0001, in comparison to si-NC group with TNF-α treatment (**D**–**F**).

**Figure 6 pharmaceuticals-16-01440-f006:**
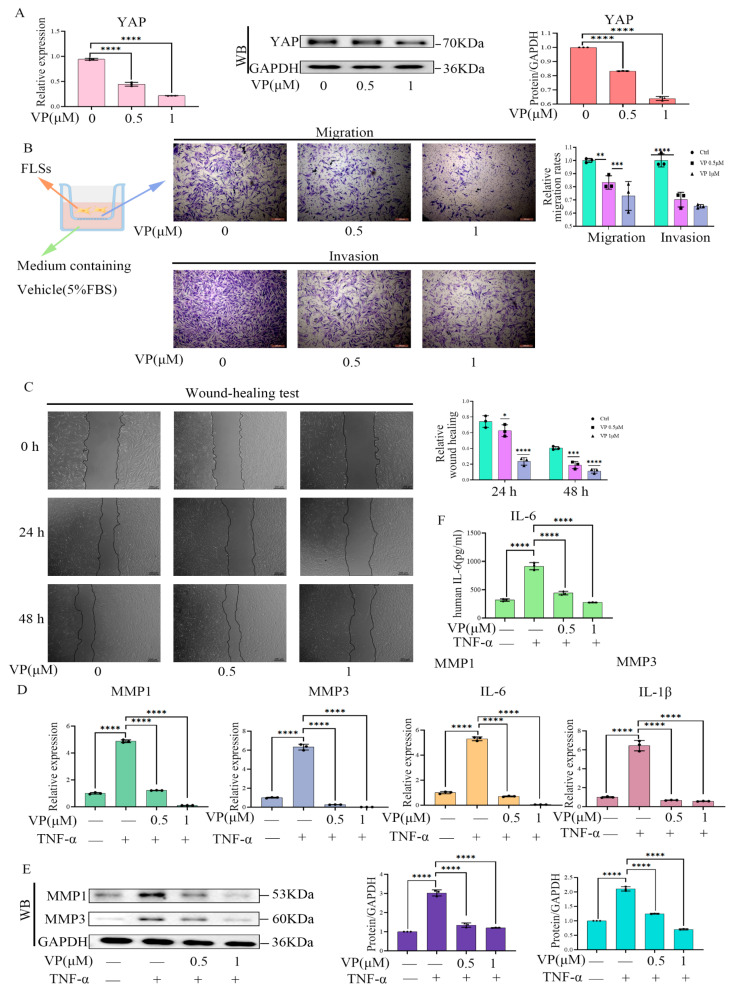
VP treatment resulted in reduced inflammatory functions of RA-FLS cells. (**A**) Western blot and qPCR were used to assess the effects of VP (0.5 and 1μM) on YAP expression in RA-FLS cells. (**B**,**C**) RA-FLS cells were stimulated with various concentrations of VP. Then, the cells were subjected to transwell assays to detect vertical migration and invasion of the cells (**B**). Wound-healing tests were performed to detect cell horizontal migration (**C**). (**D**,**F**) VP (0.5 and 1μM) effects on MMP1, MMP3, IL-6, and IL-1β expression in RA-FLSs were assessed using qPCR (D), and IL-6 in the cell culture supernatants was examined using ELISA (**F**). (**E**) Western blot analysis was conducted to detect TNF-α-induced (50 ng/mL) MMP1 and MMP3 expression in RA-FLS after VP treatment for 48 h. All experiments were conducted independently three times. The data are shown as mean ± SEM. * *p* < 0.05, ** *p* < 0.01, *** *p* < 0.001, **** *p* < 0.0001, in comparison to si-NC RA-FLS group (**A**–**C**). **** *p* < 0.0001, in comparison to si-NC group with TNF-α treatment (**D**–**F**).

**Figure 7 pharmaceuticals-16-01440-f007:**
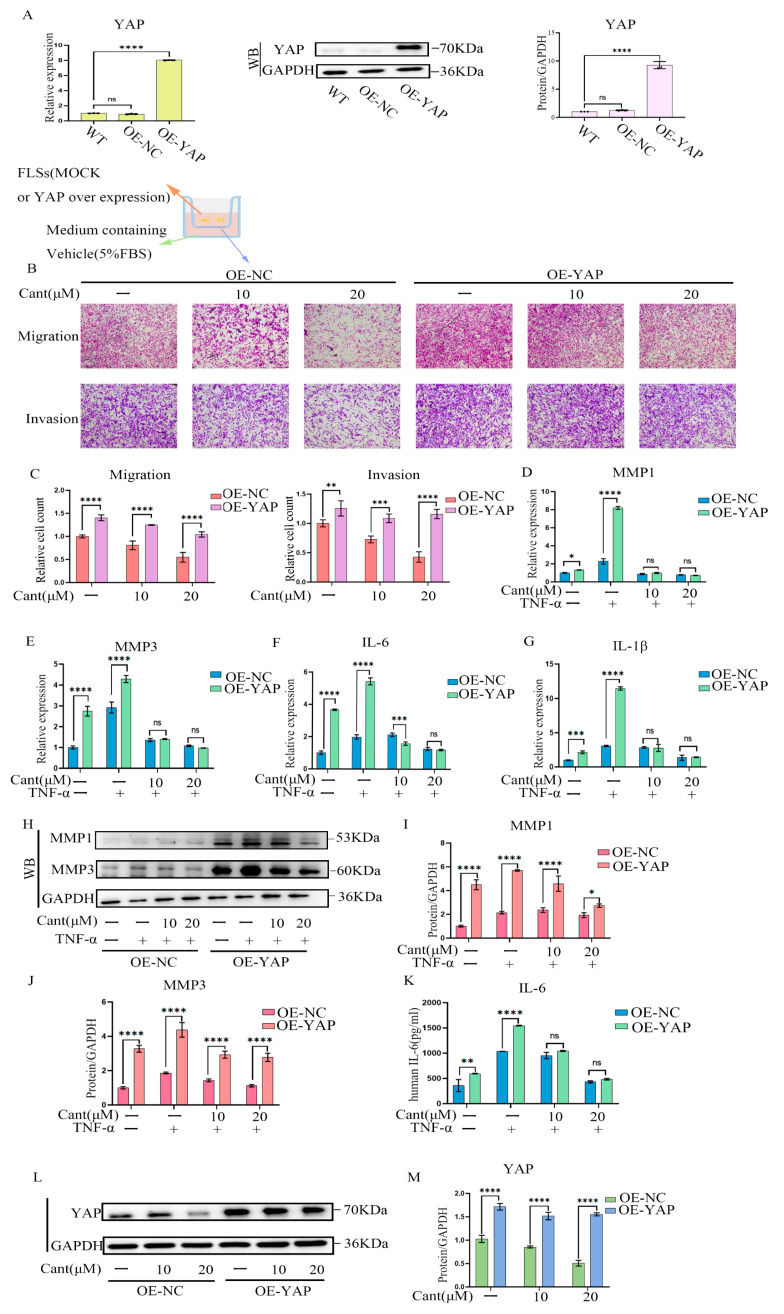
YAP over-expression significantly reversed Cant-induced decline in RA-FLS cell migration and invasion. (**A**) The expression of YAP in RA-FLS cells transfected with a lentiviral vector carrying YAP (OE-YAP) or an empty vector (OE-NC) were measured by qPCR and Western blot assay. (**B**,**C**) Transwell assays showed the capacity of vertical migration and invasion of Cant-treated OE-NC and OE-YAP RA-FLS cells. (**D**–**G**) The effects of Cant on the expression of MMP1, MMP3, IL-6, and IL-1β in OE-NC and OE-YAP RA-FLS cells were measured by qPCR. (**H**–**J**) The OE-NC and OE-YAP cells were pre-treated with Cant for 1 h and stimulated with TNF-α (50 ng/mL) for 24 h. Then, the expressions of MMP1 and MMP3 were analyzed by Western blot. (**K**) IL-6 in the cell culture supernatant was tested by ELISA. (**L**,**M**) Western blot detection of proteins in OE-NC and OE-YAP RA-FLS cells after 24 h of Cant stimulation. All experiments were performed independently three times, and data are shown as mean ± SEM. * *p* < 0.05, ** *p* < 0.01, *** *p* < 0.001, **** *p* < 0.0001, compared with OE-NC RA-FLS group.

**Figure 8 pharmaceuticals-16-01440-f008:**
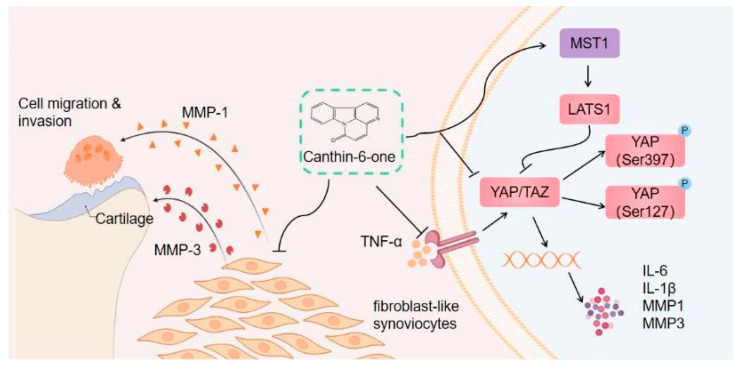
Proposed mechanism of canthin-6-one on inhibition of RA-FLS migration and invasion. Cant limits synovial fibroblast migration and invasion and reduces joint damage by up-regulating the expression of Hippo pathway regulators, MST1 and LATS1, thus down-regulating the expression and activation of YAP (indicated by phosphorylation of YAP on Ser397 and Ser127) and TAZ, which results in reduced production of TNF-α-induced MMPs (including MMP1 and MMP3) and pro-inflammatory cytokines (IL-6 and IL-1β).

**Table 1 pharmaceuticals-16-01440-t001:** Sequences of primers used in qRT-PCR. F: Forward; R: Reverse.

Genes	Sequences	
*GAPDH*	F:	CCACTCCTCCACCTTTGACGC
	R:	CCACCCTGTTGCTGTAGCCA
*MMP1*	F:	TTTGTCAGGGGAGATCATCGG
	R:	TCCAAGAGAATGGCCGAGTT
*MMP3*	F:	TGGACAAAGGATACAACAGGGAC
	R:	ATCTTGAGACAGGCGGAACC
*IL-6*	F:	TTCTCCACAAGCGCCTTC
	R:	AGAGGTGAGTGGCTGTCTGT
*IL-1β*	F:	AACCTCTTCGAGGCACAAGG
	R:	GTCCTGGAAGGAGCACTTCAT

## Data Availability

Data is contained within the article.

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
