# Peer review of "Methyl Canthin-6-one-2-carboxylate Restrains the Migration/Invasion Properties of Fibroblast-like Synoviocytes by Suppressing the Hippo/YAP Signaling Pathway"

_pharmaceuticals, 2023, doi:10.3390/ph16101440_

Round 1

Reviewer 1 Report

Concerning the Manuscript; pharmaceuticals-2597501-peer-review-v1

Methyl Canthin-6-one-2-carboxylate Inhibits the Migration and 2 Invasion of Fibroblast-like Synoviocytes by Suppressing the 3 Hippo/YAP signaling pathway

I have read the manuscript with interest, it is written well, the selection of references is correct but minor issues/information should be addressed that I hope helps to improve this manuscript before the publication in Pharmaceuticals.

1.      The authors should be focused on the novelty of this study.

2.      The catalog number and company name for all kits used in the study should be added.

3.      Did you design the used primers? If you did, you need to mention that, otherwise you need to cite a reference.

4.   Why the author selects these concentration (10Μm / 20μM) of Cant? On what basis were these concentrations selected?

Reviewer 2 Report

The manuscript by Zhang et al. demonstrates that a canthin-6-one alkaloid, Cant, suppresses the migration and invasion of fibroblast-like synoviocytes (FLS) in vitro, which may potentially be developed as a therapeutic for rheumatoid arthritis (RA). FLS plays a crucial role in the pathogenesis of RA, and developing therapeutics to modulate the function of FLS may be an effective therapeutic option for RA. While the study is novel and the manuscript is easy to follow, the major issue is that the authors do not discuss their findings in detail or provide interpretations. Additionally, Cant can have multiple targets, which could lead to side-effects. The study also has inconsistencies listed below.

·       The figure legend for Figure 1 mentions that Cant did not affect apoptosis. However, the authors describe in their Results that Cant significantly reduced the total % of apoptotic cells. This is contradictory.

·        Figure 1C: The dot plots for Annexin V and PI are not convincing. How were the gates decided? The authors should include a positive control in this experiment. Also, which population of cells are the authors considering apoptotic: Annexin V and PI double positive or only Annexin V?

·        Figure 1E: The microscopic images are of poor quality, and the Hoechst staining is not visible in the version of the manuscript I have.

·        Was TNF-a used in the experiments described in Figure 1? The figure legend mentions this.

·        Figure 2: How and why did the authors choose only MMPs 1-3? The authors should discuss why changes in MMP2 levels were not observed.

·        Is there a reason why the authors show protein levels (Fig. 2D) first and then the transcripts (Fig. 2H)? Generally, transcripts are shown first and then the protein.

·        Figures 3B-C: Please elaborate on vertical and horizontal migration differences.

·        Figure 4: Cant can exert its effect on multiple signaling pathways. While the authors investigate YAP/TAZ pathway, it is unclear whether this is the only pathway or one of the pathways. This needs to be described in the manuscript. Moreover, YAP/TAZ pathway plays a role in apoptosis. Since the authors observed that Cant affects apoptosis, could this be the reason for a reduction in YAP/TAZ?

·        Figure 4A: If the total protein levels of YAP are reduced with Cant, the phospho-YAP will also reduce. What are the authors trying to convey here?

·        Figure 4D: Similar to the comment above, the Hoechst staining is not visible.

·   Figures 5&6: The data suggests that the YAP pathway can be targeted. But unclear as to what the authors are trying to convey with these experiments as Cant is not used. Is a YAP-specific inhibitor, like VP, better than Cant? The authors should demonstrate what happens to the expression of MMPs and IL-6 via ELISA when TNF-a-stimulated FLS are incubated with VP.

Minor typos and grammatical errors were noted.

Round 2

Reviewer 2 Report

The authors have addressed all the comments. The new data in Figure 7 strengthens the study.